# Photocatalytic Oxidation of Natural Organic Matter in Water

**Dan C. A. Gowland [1], Neil Robertson [2] and Efthalia Chatzisymeon [1,\*]**

[1] Institute for Infrastructure and Environment, School of Engineering, The University of Edinburgh, Edinburgh EH9 3JL, UK; d.c.a.gowland@sms.ed.ac.uk

[2] EaStCHEM School of Chemistry, Joseph Black Building, King's Buildings, Edinburgh EH9 3FJ, UK; neil.robertson@ed.ac.uk

\* Correspondence: e.chatzisymeon@ed.ac.uk; Tel.: +44-(0)-1316505711

**Abstract:** Increased concentrations of natural organic matter (NOM), a complex mixture of organic substances found in most surface waters, have recently emerged as a substantial environmental issue. NOM has a significant variety of molecular and chemical properties, which in combination with its varying concentrations both geographically and seasonally, introduce the opportunity for an array of interactions with the environment. Due to an observable increase in amounts of NOM in water treatment supply sources, an improved effort to remove naturally-occurring organics from drinking water supplies, as well as from municipal wastewater effluents, is required to continue the development of highly efficient and versatile water treatment technologies. Photocatalysis has received increasing interest from around the world, especially during the last decade, as several investigated processes have been regularly reported to be amongst the best performing water treatment technologies to remove NOM from drinking water supplies and mitigate the formation of disinfection by products. Consequently, this overview highlights recent research and developments on the application of photocatalysis to degrade NOM by means of $TiO_2$-based heterogeneous and homogeneous photocatalysts. Analytical techniques to quantify NOM in water and hybrid photocatalytic processes are also reviewed and discussed.

**Keywords:** DBPs; AOPs; advanced oxidation processes; fulvic acid; humic acid; wastewater treatment

## 1. Introduction

Natural organic matter (NOM) poses a significant threat to the treatment of drinking water by adding several complications to standard processing methods as well as presenting a substantial risk to public health. NOM is a complex matrix of organic compounds mostly made up of a mixture of humic and fulvic substances including anionic macromolecules of various molecular weights with both aromatic and aliphatic components. Humic acids are mostly made up of larger (10,000 to 100,000 Da) alkaline soluble molecules that vary greatly on the source of material they decay from whereas fulvic acids (fulvates, molecules bound to minerals, and free-form fulvic acids) are usually smaller than humic acids (1000 to 10,000 Da) and are soluble at most pH levels. NOM levels found in most natural waters range from 0.1 to 20 mg/L, [1–3] however an increase in its concentration in environmental water matrices has been observed recently, [4–6] presenting a strain on current water treatment infrastructure and local ecosystems. This increase in NOM concentration can be attributed to several drastic changes to climate conditions [7–9]. For example, there is a correlation between intensity of precipitation and NOM concentration discharged from forested sites, giving rise to increased runoff intensities and therefore, increased discharge from soils rich in soil organic matter (SOM). Decreased retention time in lake waters, due to climate change induced precipitation, may further cause decreased photochemical degradation of coloured NOM, decreased microbial degradation of complex organic compounds, and decreased in-lake NOM coagulation and sedimentation [10]. Additionally, apparent changes in colour and UV absorption relative to total organic carbon

(TOC) [11] also imply a change in NOM characteristics and therefore treatability meaning diversification of NOM removal is needed now more than ever [12]. High NOM concentrations can cause aesthetic problems, such as colour and taste [13] in drinking water, as well as higher maintenance and treatment costs of water and wastewater [14,15]. Most importantly, recent studies show that certain classes of NOM can react with chemicals (e.g., chlorine) used in the water treatment process, leading to the formation of carcinogenic disinfection by-products (DBPs) and trihalomethanes (THMs) [16–18]. Since the discovery of DBP formation, several studies have outlined associations between consumption of chlorinated tap water containing elevated THM concentrations and adverse health outcomes, including bladder cancer, [19] children born small for gestational age, [20,21] and miscarriages [22]. Another adverse effect indirectly caused by the presence of NOM in surface waters is the observed interference humic substances have on water treatment processes that are targeting toxic compounds or heavy metals. For example, there has been a significant amount of investigation on the inhibitory effects of NOM on targeted wastewater treatments for residual pharmaceuticals which has been shown to significantly decrease the efficiency of such processes [23–28].

Current alternative treatment techniques for NOM removal, such as coagulation, [29] adsorption, [30] membrane filtration, [31,32] flotation, [33] biological, [15] and ion exchange (IE) [34] processes also bring their own set of problems. For example, the pre-treatment for micro/ultrafiltration systems using conventional treatment processes such as coagulation/flocculation which can partially remove NOM, show low removal efficiency at lower NOM concentrations. Nanofiltration is also sometimes used as a method of NOM removal but additionally comes with the problem of significant membrane fouling [35]. These problems all show a clear need for an alternative method of removing NOM from water resources.

Advanced oxidation processes (AOPs) are widely applied methods for removal of NOM and water treatment [36]. Within the area of AOPs, photocatalysis is an up-and-coming area of research due to its, until recently, untapped wide potential for possible environmental engineering applications. Ongoing research on photocatalytic NOM removal is based around the use of semiconductors (e.g., $TiO_2$ and ZnO) as sensitizers for light-induced redox processes. When illuminated with a photon of energy greater than the bandgap energy, these semiconductors form an electron/hole pair. These electron/hole pairs are powerful redox species which many organic photodegradation reactions utilize either directly or indirectly via formation of hydroxyl radicals in solution, [37,38] as shown in Figure 1. Early research tested the capabilities of these reactions using low efficiency UV lamps as $TiO_2$, the most commonly used photocatalyst, has a fairly low visible light absorption. Whereas current work has shifted over to the use of solar and high efficiency light emitting diodes (LEDs) as sustainable photocatalytic irradiation sources [39–44].

Photocatalysis is commonly categorised into either heterogeneous or homogeneous depending on whether the catalyst is in a different phase from the reactants (heterogeneous) or in the same phase (homogeneous). Most common heterogeneous photocatalysts are transition metal oxides and semiconductors, $TiO_2$ being the most researched due to its high photocatalytic activity, excellent physical and chemical stability, low cost, and nontoxicity to humans and the environment. Other common heterogeneous photocatalysts include zinc oxide (ZnO), which also shows great photocatalytic activity, [45–48] and graphitic carbon nitride (g-$C_3N_4$), which is being increasingly used because of its preferable bandgap for visible light reactions [49]. Heterogeneous photocatalysis gives practical advantages as it allows easy separation of the reaction media from the catalyst as well as high levels of chemical stability and reusability with many new compounds being developed each day [50–52]. Alternatively, homogeneous photocatalysis may require more complicated steps for catalyst removal but has shown very high photocatalytic activity. The most commonly used homogeneous systems are based on the photo-Fenton process ($Fe^{2+}/H_2O_2$) where the hydroxy radicals produced are the reactive species [53–56].

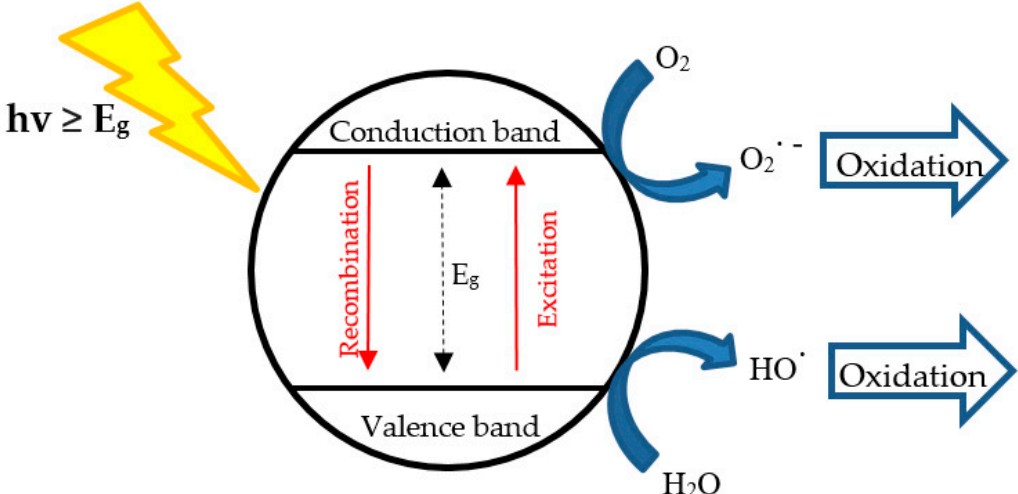

**Figure 1.** Principle mechanism of photocatalysis.

Increased NOM concentrations in aqueous environment and their adverse effects on human health and ecosystems have been extensively reported. In addition, there are several publications demonstrating that photocatalytic oxidation is a very promising process to remove NOM from water [57]. Nevertheless, a systematic review of all these studies that will be able to summarize all previous research findings, highlight important issues and research areas for further study, as well as suggest new ways to increase the effectiveness and sustainability of existing practices in water and wastewater treatment plants is still missing from literature. Therefore, the aim of this study is to provide a comprehensive review of the work surrounding the photocatalytic treatment and removal of NOM in water resources. Publications on $TiO_2$-based heterogeneous and homogeneous photocatalytic oxidation are systematically presented and discussed. Recommendations for future research directions and approaches that show promise in advancing these areas are made.

## 2. Materials and Methods

"Natural organic matter", "water", and "wastewater" were used as topic words in searching for papers and patents in Web of Science, Scopus, and Google Scholar (as supplementary) without restriction on publication date. Related documents (>100) were selected to summarize research findings on NOM treatment using photocatalytic degradation as well as other current methods for NOM treatment.

## 3. Results and Discussion

### 3.1. Analytical Techniques to Detect and Quantify NOM in Water

The type and amount of NOM in water substantially varies among different environmental matrices, as it strongly depends on climatic conditions, hydrological regime as well as other environmental factors. Therefore, to monitor NOM concentration during treatment at lab-scale and improve reproducibility of results, synthetic NOM solutions are commonly used. Common procedures involve dissolving humic and/or fulvic acids in ultrapure water and using them to mimic NOM, as humic and fulvic acids represent up to 80% of the dissolved organics in natural waters and have been shown to be DBP precursors. The reduction of DBP from drinking water is the subject of many NOM related projects, therefore simple, refined humic and fulvic acid samples that are commercially available are typically used by researchers, at least in preliminary testing [58]. More complex NOM samples extracted from water (e.g., International Humic Substances Society (IHSS) samples from the Suwannee River and Mississippi River) have also been known to be used to simulate surface waters as they can give results that more accurately reflect the conditions

of certain waters in a controlled manner, although they are more expensive than simpler synthetic NOM varieties [59].

Different methods are applied to quantify NOM in water: Specific UV absorbance at $\lambda$ = 254 nm (SUVA), [60–63] chemical oxygen demand (COD), [64] total organic carbon analyser (TOC), [65,66] fluorescence spectroscopy, [67–70] high performance liquid chromatography (HPLC), [71,72] and mass spectrometric methods (MS) [73,74]. The pros and cons of these commonly used methods are weighed up in Table 1. UV-Vis spectroscopy, in the range of 254–280 nm, is commonly used to measure the presence of unsaturated double bonds and $\pi$–$\pi$ electron interactions typically found in aromatic compounds such as humic acid. Measuring SUVA is fast, simple, and does not require complicated equipment or chemical reagents making it a popular analytical method in NOM related research. COD utilises an oxidising agent to measure the amount of oxygen needed to oxidise organic matter in solution (permanganate index). This method has been used for a long time due to its simplicity but the many hazardous chemicals (i.e., mercury, hexavalent chromium, sulfuric acid, and silver) involved in the process mean more complex waste management is required than most other methods. Standard COD methods also only allow for COD concentrations that are >50 mg/L with alterations to allow for COD detection from 5 to 50 mg/L [75] making its applications in wastewater management very limited. TOC analysis, considered the main indicator of NOM in the drinking water, determines the organic carbon present in solution by using infrared (IR) spectroscopy to measure the carbon dioxide produced by heat catalysed chemical oxidation with a persulfate solution. Although, compared to UV254 spectroscopy, TOC analysis requires a slightly longer runtime and a more complex preparation, it is still a relatively quick and simple method for quantifying NOM in water with many different available conditions to allow for the tailoring of specific test environments. The chemicals generally required in TOC also have low toxicity and are cheaply available compared to other methods available. Other analytical methods to detect and quantify NOM are also widely used across various disciplines. The complexity of these methods is increased as more information about the NOM's profile is afforded [76–82].

### 3.2. NOM Photocatalytic Treatment

### 3.2.1. Heterogeneous TiO₂ Photocatalysis

Table 2 shows all publications focusing on treatment of NOM in water by means of heterogeneous TiO₂ photocatalysis. Some of the earliest work on the photocatalytic degradation of NOM was done by Bekbölet et al. [83,84] where slurries of P25-TiO₂ were used to explore the limitations and general trends observed when changing the reaction conditions in standard photocatalytic procedures. Although these early papers reported on the most favourable conditions for photocatalytic NOM removal, much more work has been done since on optimizing the resultant degradation of NOM by altering basic operational parameters.

**Table 1.** Table presenting the main analytical techniques for natural organic matter (NOM) detection and quantification in water.

| Method | Advantages | Disadvantages | Complexity of Method |
|---|---|---|---|
| Adsorption at 254 nm | — Ease of use <br> — Very fast measurement <br> — Cheap | — Measurement of unsaturated organics in water (not only NOM/humic acid) <br> — High nitrate content in low dissolved organic carbon (DOC) waters may interfere the measurement | Low |
| COD | — Simple <br> — Well known method | — Toxic treatment chemicals <br> — Low accuracy <br> — High minimum detection limit | |
| TOC | — Fast measurement <br> — Tailorable modes of detection for specific experiments | — Expensive specialised equipment <br> — Measurement of total organics in water (not only NOM/humic acid) | |
| Fluorescence spectroscopy | — No pre-treatment required <br> — Gives information on specific NOM | — Only detects fluorescent NOM molecules <br> — Sensitive to chemical environment, e.g., pH | |
| FTIR | — Good signal to noise ratio <br> — Extensive libraries of humic substances to identify specific compound characteristics | — Can see large water band interference <br> — Pre-treatment could alter chemical makeup of NOM | |
| HPLC | — Good separation of NOM compounds | — Requires expensive, high purity solvents, columns etc. <br> — NOM can have unwanted interactions with the stationary phase | |
| GC-MS | — Accurate detection of all substances found in water | — Cost of reagents, columns, etc. <br> — Difficulty in analysing and interpreting results | High |

Table 2. Publications on heterogeneous photocatalytic treatment of NOM.

| Water Matrix | Catalyst Type | Reaction Time | Irradiation Source | Other Operating Parameters | Removal Efficiency | Other Important Findings | Reference |
|---|---|---|---|---|---|---|---|
| Humic acid solution | P25-TiO$_2$ | 120 min | UVA—125 W | Ambient pH [HA] = 50 mg/L [TiO$_2$] = 1 g/L | 88% TOC 99% Vis$_{400}$ | THMFP * = 14.5 µg/L | Bekbölet et al. (1996) [83] |
| Humic acid solution | P25-TiO$_2$, UV100-TiO$_2$ | 60 min | UVA—125 W $\lambda$ = 300–420 nm | Ambient pH [HA] = 10 mg/L [TiO$_2$] = 0.25 g/L | P25: 70% TOC UV100: 50% TOC | NOM removal rate constant: P25 = 1.9 × 10$^{-2}$ min$^{-1}$ UV100 = 1.2 × 10$^{-2}$ min$^{-1}$ | Bekbölet et al. (2002) [84] |
| Reservoir water: M-Myponga site W-Woronora site | P25-TiO$_2$ | 150 min | UVA—20 W $\lambda$ = 365 nm | pH~7 TOC$_M$ = 10.6 mg/L TOC$_W$ = 3.5 mg/L [TiO$_2$] = 0.1 g/L | M: 80% TOC 100% UV$_{254}$ W: 80% TOC 100% UV$_{254}$ | THMFP: M = < 20 µg/L W = < 20 µg/L | Liu et al. (2010) [85] |
| Sand filtered treatment plant water | N-Pd-TiO$_2$ | 120 min | Solar simulator 500 W | pH~6.73 TOC = 2.38 mg/L [N-Pd-TiO$_2$] = 5 g/L | HPO ** = 71% HPI ** = 35% TPI ** = 15% UV$_{254}$ | | Nkambule et al. (2012) [86] |
| Reverse osmosis isolate and Alginic acid solution | AgSiO$_2$-TiO$_2$ | 30 min | Solar simulator $\lambda$ = 400–1100 nm | pH~8.2 TOC$_I$ = 3.7 mg/L [TiO$_2$] = 0.1 g/L | 20% TOC 42% UV$_{254}$ | 219 ± 40 µg THMFP per g TiO$_2$ | Gora et al. (2018) [87] |
| Humic acid solution | Al:Fe-TiO$_2$ (1%) | 15 min | UVC—37 W $\lambda$ = 254 nm | pH~7 [HA] = 10 mg/L [TiO$_2$] = 0.1 g/L O$_3$ | 63.2% TOC 79.4% UV$_{254}$ | Increasing HCO3- concentration decrease NOM reduction rate | Yuan et al. (2013) [88] |
| Reservoir water: MV-Midvaal P-Plettenberg bay | MWCNT/N, Pd-TiO$_2$ *** | 120 min | Solar simulator 300 W | [MWCNT/N, Pd-TiO$_2$] = 1 g/L | MV: 69.4% P: 97.7% UV$_{254}$ | | Ndlangamandla et al. (2018) [89] |
| Humic acid solution | TiO$_2$ nanotubes | 120 min | UVC—11 W $\lambda$ = 254 nm | [HA] = 50 mg/L [TiO$_2$] = 0.5 g/L | 98.27% DOC 100% UV$_{436}$ | Humic acid removal rate: 0.0607 molm$^{-3}$s$^{-1}$ | Zhang et al. (2009) [90] |
| Landscape surface water | Bi$_2$O$_3$-TiO$_2$ | 10 min | Vis—300 W $\lambda$ = 400–780 nm | pH~8.13 TOC$_I$ = 2.2 mg/L [Bi$_2$O$_3$-TiO$_2$] = 2 g/L | 20.2% TOC 24.4% UV$_{254}$ | | Wang et al. (2019) [91] |

**Table 2.** *Cont.*

| Water Matrix | Catalyst Type | Reaction Time | Irradiation Source | Other Operating Parameters | Removal Efficiency | Other Important Findings | Reference |
|---|---|---|---|---|---|---|---|
| Pre-treated (coagulation-flocculation) water | P25-TiO$_2$, TiO$_2$/β-SiC | 220 min | Solar simulator—1500 W | pH~6.7<br>P25:<br>TOC$_I$ = 7.8 mg/L<br>[TiO$_2$] = 0.5 g/L<br>β-SiC:<br>TOC$_I$ = 5.5 mg/L<br>[TiO$_2$] = 0.5 g/L | P25:<br>80% TOC<br>β-SiC:<br>80% TOC | | Ayekoe et al. (2017) [92] |
| Treatment plant inlet water in immersed ultrafiltration module | P25-TiO$_2$ | 120 min irradiation 43 h total treatment | UVC—15 W<br>λ = 254 nm | pH~7<br>DOC = 5.48 mg/L<br>[TiO$_2$] = 0.1 g/L | 60% DOC<br>90% UV$_{254}$ | THMFP * = 25 μg/L | Rajca et al. (2016) [93] |
| Humic acid solution | LiCl-TiO$_2$ doped PVDF **** membrane | 30 min | UVA—100 W<br>λ = 365 nm | pH~7.5<br>[HA] = 2 mg/L | 80–84% UV$_{254}$ | | Song et al. (2014) [94] |
| Extracted river NOM | P25-TiO$_2$ | 120 min | UVC—8 W<br>λ = 254 nm | pH~8.2<br>TOC$_I$ = 10 mg/L<br>[TiO$_2$] = 1 g/L | 80% TOC<br>100% UV$_{254}$ | NOM degradation rate constant: 0.0163 min$^{-1}$ | Huang et al. (2008) [95] |
| River water | Nano-TiO$_2$ on diatomite | 360 min | 3× UVC lamps—16 W<br>λ = 254 nm | pH~8.0–8.5<br>TOC$_I$ = 9.84–13.18 mg/L<br>[TiO$_2$] = 0.5 g/L | 28.5% TOC<br>40% UV$_{254}$ | | Sun et al. (2014) [96] |
| Humic acid solution | TiO$_2$ nanoparticles/granular activated carbon (GAC) | 180 min | UVA—500 W<br>λ = 365 nm | pH~4.2<br>TOC$_I$ = 5.04 mg/L<br>[TiO$_2$/GAC] = 2 g/L | 99.5% UV$_{254}$ | Significantly lower degradation (70% UV$_{254}$) at pH = 11 | Xue et al. (2011) [97] |

* THMFP—Trihalomethane formation potential. ** HPO—Hydrophobic, HPI—Hydrophilic, TPI—Transphilic. *** Multi-walled carbon nanotubes/nitrogen, palladium co-doped titanium dioxide. **** Polyvinylidene difluoride.

Catalyst and NOM Concentration

The concentration of catalyst is an important parameter for photocatalytic oxidation processes. In general, photocatalytic oxidation is enhanced when catalyst concentration is increased up to a value at which removal rate reaches a plateau. Bekbölet et al. [83] observed that an optimal catalyst concentration for NOM removal is 1 g/L and this is high enough to provide a large surface area to adsorb NOM but not so high that the slurry significantly reduces the availability of light throughout the reaction mixture. This upper limit has been noted to depend on the geometry and on the working conditions of the photoreactor [98]. This same study evaluated the trihalomethane formation potential (THMFP) of various humic acid concentrations with respect to time, which also opened up the possibility for more research on the area by showing up to 94% reductions in THMFP (below the United States Environmental Protection Agency limits) after 120 min of photocatalytic treatment. A similar experiment by Maleki et al. [47] investigated the effects of catalyst concentration on humic acid degradation using Cu-doped ZnO nanoparticles. The same characteristic rise and plateau in degradation rate was observed when increasing the catalyst concentration from 1.5 g/L to 2 g/L, which was attributed to partial catalyst agglomeration and a consequent decrease in the active catalyst surface at higher dosages of catalysts. Additionally, early work from Palmer et al. [99] on the operational conditions of photocatalytic NOM degradation using $TiO_2$ showed that the rate of initial degradation increased with increasing concentration until the concentration of 30 ppm of carbon, after which, the rate decreased. This trend is seen in several other studies of NOM degradation above 30 ppm of C [100–102].

UV-Light Driven $TiO_2$ Catalysts

Another early paper by Bekbölet et al. [84] investigated the differences in performance between; the standard $P25-TiO_2$, a 20:80 mix of rutile: Anatase titania, and another commercially available HOMBIKAT UV 100-$TiO_2$, made entirely of anatase phase. Here it was shown that $P25-TiO_2$ showed better photocatalytic activity when it came to humic acid degradation. This correlated with the adsorption experiments also performed, which revealed that the P25 adsorbed three times as much humic acid at a set loading than the UV100 despite having a much lower BET surface area. Due to its proven high-performance rate $P25-TiO_2$ is often the comparative standard used in NOM photocatalytic degradation studies of various semiconductor loadings. Huang et al. [95] in 2008 studied the photocatalysis' effect on NOM by analysing its compositions in water before and after irradiation. There was observed preferential removal of high molecular weight, hydrophobic NOM molecules, which are the major NOM fraction responsible for membrane fouling [85,86]. Further research by Valencia et al. [103] using size-exclusion chromatography with respect to operating pH levels during photocatalytic degradation showed that changes in pH values affected the adsorption of NOM onto $TiO_2$. It was established that the pH determines the mechanism, but not the sequence of the photocatalytic degradation and therefore, regardless of pH, the degradation of the NOM preferentially removed the larger molecular size fraction in comparison to the middle and small fractions. Other comprehensive work on operating parameters includes that done by Espinoza et al. [104] on the effect of metal ions ($Cu^{2+}$, $Fe^{3+}$, $Zn^{2+}$, and $Mn^{2+}$) in solution on the photocatalytic degradation of NOM. Photocatalytic degradation experiments with and without the addition of various combinations of $CuSO_4$ or $CuCl_2 \cdot 2H_2O$, $FeCl_3$, $ZnCl_2$, and $MnCl_2$ solutions revealed an apparent reduction in photoactivity and prevention of certain degradation products when in the presence of added $Cu^{2+}$ ions (10 μM). The addition of $Mn^{2+}$ was observed to change the magnitude of the effect of added $Cu^{2+}$ a larger inhibiting effect from added $Cu^{2+}$ was observed in the absence of added $Mn^{2+}$ during the degradation of large molecular weight NOM. It was suggested by Espinoza et al. that these observations could be explained by a stabilization of the NOM against degradation by HO· by complexation with $Cu^{2+}$, which would increase the longevity of NOM in aquatic systems. Adding $Fe^{3+}$ and $Zn^{2+}$ to the experiments showed no significant effects.

Visible Light Driven Modified TiO$_2$ Catalysts

Many investigations on photocatalytic reactions are performed under UV light due to the band gap energy of pure TiO$_2$ (3 or 3.2 eV in rutile or anatase phase), which means that there is very limited photocatalytic activity in the visible range. On top of this, unmodified TiO$_2$ can be characterised with a high recombination rate for the photo-produced electron and hole pairs, and a significant difficulty to strongly bind to some support materials [105]. Therefore, a significant amount of research surrounding the photocatalytic degradation of NOM is focussed on improving the photo-efficiency of TiO$_2$ and its degradation efficiency of organic compounds. Various approaches to do so consist of chemical and structural modification of TiO$_2$, in order to enable light absorption in the visible region. These studies include several chemical modification schemes that report promising options to improve photocatalytic activity. Chemical modifications to TiO$_2$ involve the addition of various other species, typically involving: Metals (such as Fe, Pd, or Ag) [86,88,106,107]/metal oxides (such as Bi$_2$O$_3$) [108–111] which can either facilitate electron–hole separation and promote interfacial electron transfer or decrease the TiO$_2$ band gap. This promotes electron transfer from the valence band to the conduction band, facilitating the formation of oxidative species such as hydroxyl radicals [112].

Other chemical modifications include the addition of non-metals (such as C, N, S, or F) which have also been shown to form new impurity levels close to the valence or conduction band of TiO$_2$, thereby lowering the optical gap and shifting the absorption edge to the visible light region [113]. Nkambules [86] 2012 work focuses on N-doped TiO$_2$, a growing area of photocatalysis which has been shown to increase visible light photocatalytic activity when coupled with co-dopant metals by reducing the band gap of TiO$_2$ and shifted the absorption into the visible light region [89,114]. The Pd-modified N-doped TiO$_2$ catalyst synthesised by Nkambule et al. in 2012 showed a particularly successful shift in titania's visible light absorption with an over 70% removal in hydrophobic NOM fractions using a solar simulator. A drawback to these N-doped TiO$_2$ species would be that the doping of N into the lattice of TiO$_2$ usually results in the formation of oxygen vacancies in the bulk material [115]. These defects can act as recombination centres for carriers and therefore, compared to pure TiO$_2$, a loss of UV-activity is usually found for N-doped TiO$_2$, which is due to the rapid recombination rate of generated electrons and holes introduced by the impurity level. The addition of non-metals to metal doped TiO$_2$ can also be utilised to improve the stability of the photocatalyst, for example forming a thin layer of SiO$_2$ around a catalyst's surface to prevent oxidation of metal nanoparticles like Gora et al. in their 2018 investigation on modified TiO$_2$ for solar photocatalysis [87]. This work saw a Ag-TiO$_2$ nanoparticle co-catalyst reduce NOM levels by 42% UV$_{254}$ only 30 min of treatment time. This study also found significant changes to the disinfectant by-product formation potential (DBPFP) of NOM wherein the different modifications to TiO$_2$ followed the same trend in DBPFP level changes but by differing amounts.

Immobilized Catalysts

Alongside the chemical changes, various nano structured TiO$_2$ materials have been tested with enhanced visible light photoactivity such as nanoparticles, [116] nanotubes, [90] nanowires, [117] and nanofilms [118]. As well as affecting the photoactivity of TiO$_2$, many of these structurally modified materials combat the problems faced by loose slurry reactions such as catalyst separation, recovery, and reuse which bring about significant obstacles for practical applications of TiO$_2$ powder heterogeneous photocatalysis due to its small particle size [119]. Many researchers apply membrane filtration for the separation of nanosized TiO$_2$ from treated water however, as mentioned previously, serious membrane fouling usually occurs as the TiO$_2$ forms a cake layer and blocks membrane pores. Work done by Zhang et al. [90] showed that creating titania nanotubes can not only improve upon P25-TiO$_2$'s ability to photocatalytically degrade NOM, most likely due to increased surface area, but also significantly reduces the amount of membrane fouling caused by catalyst separation. Another approach to reducing the need for catalyst separation when

photocatalytically degrading NOM is producing hybrid materials by combining $TiO_2$ with carbon materials such as multiwalled carbon nanotubes [89] and activated carbons [97]. A particularly successful example of this is the work done by Xue et al. where a $TiO_2$ nanoparticle/granular activated carbon composite (GAC) was prepared by a sol-dipping–gel process. This investigation displayed a synergetic relationship between adsorption upon GAC and degradation involving $TiO_2$ where a humic acid removal of 99.5% $UV_{254}$ was achieved as well as improved filterability. Hybrid membranes combining $TiO_2$ with various polymeric compounds such as polyvinylalcohol, pyrrolidone [120] and poly (amide–imide) [121] also show a solution to catalyst separation. These hybrid membranes exhibit great potential for water treatment since they combine filtration and photo degradation in a single unit. Although blending photocatalytic nanoparticles into polymeric thin film can cause the entrapped photocatalyst to show reduced catalytic activity due to the agglomeration and shielding effects in the polymer matrix [122].

Hybrid Processes

The most widely used processes for the removal of NOM from water sources are separation and purification technologies including (micro, ultra, and nano) membrane filtration, reverse or forward osmosis, and coagulation. One approach to improving the overall efficiency of water treatment facilities is combing one or more of these technologies with heterogeneous photocatalysis. This includes the combining of photocatalysts with membrane filtration [93–95,120,123–125] and adsorption, [126] as well as coagulation systems [91]. An example of such systems would be Wang et al. [91] whose work, which showed pre-treatment by photocatalysis with $Bi_2O_3$-$TiO_2$ (4%), could improve the removal of organic matter compared to polyaluminium chloride (PACl) coagulation treatment alone. This study saw removal rates of 20.2% and 24.4% $UV_{254}$ after just 10 min of photocatalytic treatment which increased to 37.93% TOC and 58.75% $UV_{254}$. Photocatalytic oxidation prior to coagulation has been observed to decrease coagulation efficiency by 15%, most likely because the oxidation changes the characteristics of NOM and degrades NOM molecules towards smaller molecular mass fractions [127]. However, when oxidation was performed after coagulation, about 32% DOC and 33% $UV_{254}$ enhancements to the removal of NOM occurred [128].

These hybrid processes can also work in tandem to help reduce the inherent downfalls of photocatalytic systems. For example, to ensure an efficient rate of photocatalytic reaction, it is recommended that water turbidity should not exceed 5 NTU [129,130]. Although it has been observed that the 5 NTU limit is subjective and differs for each water source and desired treatment level [131]. This limitation on photocatalytic efficiency set by water turbidity means that conventional treatments (i.e., sieving, filtration, sedimentation, coagulation, and flocculation) may be an appropriate industrial pre-treatment for many photocatalytic processes.

3.2.2. Homogeneous Photocatalysis
Photo-Fenton

Recent interest in homogeneous photocatalytic NOM removal has increased due to reports of lower chemical doses, and therefore, lower residual levels of chemicals post treatment, when compared to conventional NOM removal methods such as coagulation [132]. Although heterogeneous photocatalysis, such as a standard $TiO_2$/UV NOM removal, mentioned previously, has the added benefit of easy separation after treatment is completed, homogeneous photocatalysis reactions have the advantage of providing a greater degree of interaction between the catalyst and the specified target due to the increased accessibility of the catalytic sites whilst in solution. The homogeneous photocatalytic degradative removal of organic compounds from water is mainly based on the generation of high amounts of hydroxyl radicals from either ozone or hydrogen peroxide. These generated hydroxyl radicals can degrade the organic matter commonly through hydrogen abstraction from aliphatic carbon atoms and electrophilic addition to double bonds or aromatic rings [133].

This is ideal for the degradation of the large hydrophobic NOM fractions which are major precursors for DBP formation [63]. Selected publications on homogeneous photocatalytic NOM treatment are highlighted in Table 3 below.

**Table 3.** Publications on homogeneous photocatalytic treatment of NOM.

| Homogeneous Processes | Water Matrix | Catalyst Type | Reaction Time | Irradiation Source | Other Operating Parameters | Removal Efficiency | Reference |
|---|---|---|---|---|---|---|---|
| Hybrid Photolysis | Reservoir water | $O_3$/UV | 60 min | UVA lamp Intensity = 9.7 mW/cm$^2$ | pH~6.6 TOC = 1.8 mg/L $O_3$ dosage = 0.62 g/L | 50% TOC | Chin and Bérubé (2005) [134] |
| | River water | $H_2O_2$/UV $O_3$/UV | 30 min | UVA lamp—43 W | TOC = 3.1 mg/L [$H_2O_2$] = 23 mg/L $O_3$ dosage = 4 mg/L | $H_2O_2$ only: 3–23% DOC 60% UV$_{254}$ $O_3$ only: 31% TOC 88% UV$_{254}$ | Lamsal et al. (2011) [135] |
| | Reservoir water | $H_2O_2$/UV | - | UVC lamp $\lambda$ = 254 nm | [$H_2O_2$] = 23 mg/L | - | Toor et al. (2005) [136] |
| Photo-Fenton | Inlet water to water treatment works | $FeSO_4 \cdot 7H_2O$ + $H_2O_2$ | 20 min | 4× UVA lamps—25 W $\lambda$ = 365 nm | pH~4 DOC = 9.6 mg/L [Cat] = 5.65 mg/L $H_2O_2$:$Fe^{2+}$ = 5:1 | 90% DOC 95% UV$_{254}$ | Murray et al. (2002) [132] |
| | Water treatment works reservoir water | $FeSO_4 \cdot 7H_2O$ + $H_2O_2$ | 30 min | 4× UVA lamps—25 W $\lambda$ = 365 nm | pH~4 DOC = 7.5 mg/L [$Fe^{2+}$] = 0.1 mM $H_2O_2$:$Fe^{2+}$ = 5:1 | 90% DOC 95% UV$_{254}$ | Murray et al. (2004) [137] |
| | Reservoir water | $FeSO_4 \cdot 7H_2O$ + $H_2O_2$ $H_2O_2$ only | 1 min | 4× UVC lamp – 12 W $\lambda$ = 254 nm | pH~4.5 DOC = 17.4 mg/L [$H_2O_2$] = 2.0 mM $H_2O_2$:$Fe^{2+}$ = 4:1 | $Fe_2SO_4 \cdot 7H_2O$ + $H_2O_2$: 88% DOC 31% UV$_{254}$ $H_2O_2$: 78% DOC 94% UV$_{254}$ | Goslan et al. (2006) [138] |
| | River water pre-treated with slow sand filtration | $FeCl_3 \cdot 7H_2O$ + $H_2O_2$ | After 6.5 KJ/L of solar energy | Solar CPC | pH~5 DOC = 2.7–3.1 mg/L [$H_2O_2$] = 20 mg/L [$Fe^{3+}$] = 1 mg/L | 90% DOC 95% UV$_{254}$ | Moncayo-Lasso et al. (2008) [139] |
| | River water | $FeCl_3 \cdot 7H_2O$ + $H_2O_2$ | After 20 KJ/L of solar energy | Solar CPC | pH~6.5 DOC = 5.5 mg/L [$H_2O_2$] = 10 mg/L [$Fe^{3+}$] = 0.6 mg/L | 55% DOC 75% UV$_{254}$ | Moncayo-Lasso et al. (2009) [128] |

A comparative study by Goslan et al. [129–138] showed that the addition of Ferrous sulphate increased UV/$H_2O_2$ ability to remove NOM from reservoir water by forming a photo-Fenton reaction.

$$Fe^{3+} + H_2O \rightarrow Fe\,(OH)^{2+} + H^+ \tag{1}$$

$$Fe\,(OH)^{2+} + h\nu \rightarrow Fe^{2+} + \cdot OH \tag{2}$$

$$Fe^{2+} + H_2O_2 \rightarrow Fe^{3+} + OH^- + \cdot OH \tag{3}$$

During the photo-Fenton process, in addition to Equations (2) and (3), hydroxyl radical formation can also occur via the following reactions:

$$Fe^{3+} + H_2O + h\nu \rightarrow Fe^{2+} + H^+ + \cdot OH \tag{4}$$

$$H_2O_2 + h\nu \rightarrow 2 \cdot OH \tag{5}$$

In the photo-Fenton process (Equations (1) and (2)), the Fenton reaction rates are strongly increased by irradiation with UV–vis light. The positive effect of irradiation on

the degradation rate is due to the photo-chemical regeneration of ferrous iron ($Fe^{2+}$) by photo-reduction of ferric complexes, which leads to additional ·OH generation [140–143]. The ferrous iron generated in solution reacts with $H_2O_2$ yielding a second ·OH radical and ferric ion (Equation (3)), and the cycle continues. A major advantage of the photo-Fenton reagent is that the reactions light absorption maximum wavelength is roughly 600 nm which gives a much larger absorption overlap with natural sunlight compared to many other common photocatalysts.

Although the exact mechanism used for degradation of NOM using photo-Fenton processes is not presently clear, work from Fukushima et al. [144] has shed some light on possible processes occurring during these degradation reactions. Fukushima's 2001 work on the degradation products produced from degrading several different NOM fractions in a photo-Fenton solution showed that the TOC decreased dependent on increasing irradiation time, indicating mineralisation of the HA to $CO_2$ during this process. Analysis on different molecular weight fractions of HA also suggested that the degradation of high molecular weight fractions of HA results in a lowering in molecular size during photo-Fenton processes.

Hybrid Photolytic Oxidation Processes

An interesting comparison to homogeneous photocatalysis is the work done with homogeneous hybrid photolysis for NOM treatment by enhancing the oxidative capabilities of common oxidising species (e.g., ozone and peroxides) with light. The advantages of these hybrid processes, as well as other AOPs including $O_3$/UV, $H_2O_2$/UV, and $H_2O_2$/$O_3$, was explored by Lamsal et al. [135] in 2011. This study specifically investigated the treatment process impact on the change of molecular weight distribution (MWD) and disinfection by-product formation potential (DBPFP) with the UV/ozone hybrid showing a significantly improved removal of NOM and reduced DBPFP when compared to UV or ozone treatment alone.

Hydrogen Peroxide Based Photocatalysis

Many factors decide on the optimum $H_2O_2$ dosage in UV degradation reactions. For UV/$H_2O_2$ NOM removal, the characteristics and concentration of the organic compounds can directly influence the overall mineralisation ability. The amount of hydroxyl radicals produced upon UV irradiation depends on the $H_2O_2$ concentration whilst $H_2O_2$ can also react with these radicals and inhibit hydroxyl radical evolution. Additionally, $H_2O_2$ absorbs UV energy, therefore, reducing the availability of UV photons for oxidising pollutants at higher $H_2O_2$ concentrations. Wang et al. [145] found, for the oxidation of humic acid, that the hydroxyl radical scavenging effect (the production and then combination of $HO_2$ into $H_2O_2$ and $O_2$) became significant when the $H_2O_2$ concentration was higher than 0.1% making this the optimum dosage. This study also noted that the presence of bicarbonate/carbonate species has a negative effect on NOM degradation due to causing competition for hydroxyl radicals, especially at high concentrations of $H_2O_2$.

Ozone Based Photocatalysis

Ozone can degrade NOM directly through ozonolysis which has been found to be fairly selective and relatively slow [146,147] so most NOM degradation research is focussed on increasing the generation of hydroxyl radicals from the decomposition of ozone in water. This includes the combination of ozone with UV irradiation to degrade NOM through quick, non-selective ozonation. Study results from Ratpukdi et al. [148] on the optimal operating conditions for UV/ozone hybrid photolysis procedures revealed that the mineralization rate of DOC provided by the processes tested ranked in the following order: Vacuum ultraviolet (VUV)/ozone > VUV > UV/ozone > ozone > UV. The study also showed that the highest DOC mineralisation rate and biodegradability increase was at a neutral pH 7 rather than in a basic environment (pH 9 and pH 11) with extremely basic conditions (pH 11) showing no synergistic hybrid effect from combining UV and ozone at all.

Research comparing $O_3$ NOM degradation with and without the addition of UVC shows a clear enhancement effect from UV light. Work by Chin and Bérubé [134] concluded that the combined UV/$O_3$ treatment is more effective at reducing organic constituents, as well as the DBPFP, in raw water than either the ozone or UV treatment alone. Lamsal et al. [129–135] investigated this hybrid effect further by showing how several AOP treatment processes impacted the change of MWD and DBPFP. The UV/ozone hybrid in this side-by-side study showed a significantly improved removal of NOM and reduced DBPFP when compared to UV or ozone treatment alone. It was also noted that this UV/ozone process induced a near complete alteration of the molecular weight of NOM from >900 Da to <300 Da.

### 3.2.3. Energy Efficiency of NOM Treatments

A significant area of interest surrounding UV photocatalysis is the energy consumption, and associated operating costs, of artificial lighting. The electric energy per order, $E_{EO}$, value was introduced by Bolton et al., [149] and is used to estimate the energy consumption of photocatalytic reactors. $E_{EO}$ is defined as the energy required for 90% degradation of a pollutant per cubic meter of contaminated water. $E_{EO}$ (kWh/m$^3$/order), for a batch-operated reactor, is calculated from the following Equation (6):

$$E_{EO} = \frac{P \times t \times 1000}{V \times 60 \times \log\left(C_i/C_f\right)} \tag{6}$$

where P is the electrical power of the irradiation source (kW), t is the irradiation time (min), V is the volume of the treated effluent (L), and $C_i$ and $C_f$ are the initial and final pollutant concentrations (mg L$^{-1}$), respectively. The $E_{EO}$ of selected significant publications are displayed in Table 4 to give an example of the relative energy efficiencies of various photocatalyic NOM treatments.

**Table 4.** Energy efficiency comparison of photocatalytic treatments of NOM.

| Process Type | Water Matrix | Catalyst Type | Electrical Power of the Irradiation Source (P)/kW | Reaction Time (t)/min | Volume (V)/L | TOC % | $E_{EO}$ KWh m$^{-3}$ Order$^{-1}$ | Reference |
|---|---|---|---|---|---|---|---|---|
| Heterogeneous | Humic acid solution | P25-TiO$_2$ | 0.125 | 120 | 0.05 | 88 | 5430 | Bekbolet et al. (1996) [83] |
| Heterogeneous | Reservoir water | P25-TiO$_2$ | 0.02 | 150 | 0.8 | 100 | 15,625 | Liu et al. (2010) [85] |
| Heterogeneous | Pre-treated (coagulation-flocculation) water | P25-TiO$_2$/βSiC | 1.5 | 220 | 0.1 | 80 | 78,687 | Ayekoe et al. (2017) [92] |
| Homogeneous | River water | H$_2$O$_2$/UV | 0.043 | 30 | 3 | 23 | 63,137 | Lamsal et al. (2010) [135] |
| Homogeneous | River water | O$_3$/UV | 0.043 | 30 | 3 | 31 | 44,472 | Lamsal et al. (2011) [135] |
| Homogeneous | Water treatment works reservoir water | FeSO$_4$.7H$_2$O + H$_2$O$_2$ | 0.1 | 30 | 1 | 90 | 50 | Murray et al. (2004) [137] |

An interesting observation from the data displayed in Table 4 is the significance the electrical power of the irradiation source (P) plays in the energy efficiency of a reaction. For example, for electrical power $\geq 0.125$ kW the $E_{EO}$ is at the order of $10^3$ (process types first and third as shown in Table 4), while this decreases to the order of 10 for $P \leq 0.1$ kW. This is due to the position of P on the numerator of Equation (6) which is then multiplied by 1000,

making relatively small differences in the power inputs of irradiation sources result in large disparities in $E_{EO}$. Moreover, when process types with similar P, for example types first and sixth are compared (as shown in Table 4), it can be observed that short treatment time (i.e., 30 min) is also important to keep the $E_{EO}$ at the low order of 10 KWh m$^{-3}$ order$^{-1}$. This shows the significance that recent advancements in LED technology have had for the prospects of industrial scale photocatalytic water treatment due to the drastically improved efficiency when compared to conventional mercury black lights.

## 4. Conclusions and Considerations for Future Research

The removal of NOM from drinking water presents a great challenge that will require the application of efficient and flexible water treatment technology or more likely a combination of synergistic technologies. A crucial process towards achieving this is the proper characterisation of NOM and its various fractions in order to accurately estimate their reactivity with the utilised treatment system. This procedure is critical in the selection and application of the most suitable treatment process by achieving the highest removal efficiency, the greatest reduction in disinfection by-product formation potential, and the best possible cost efficiency. Photocatalysis is highly regarded amongst NOM removal researchers due to the quick and nonselective character of the hydroxyl radicals produced during processing. This makes the measured differences of NOM in water less of an issue in photocatalysis when compared to other conventional NOM removal treatments. Although, various studies reported that photocatalysis can tend to have more impact on NOM's hydrophobic and higher MW compounds [150]. The non-specificity of hydroxy radicals can also be a disadvantage to photocatalytic methods in that the highly reactive HO can also interact with ions and other dissolved organics in waters which could reduce the overall efficiency of NOM removal. These unintentional side reactions have been observed during the removal of humic acid in the presence of bicarbonate ($HCO_3^-$) and halide ($Cl^-$ and $Br^-$) [151–153] ions.

Currently, the coupling of photocatalysis with other water treatment technologies is being investigated as a viable option to overcome the inadequacies of photocatalysis and the selected technology alone. As there is no standalone water treatment technique that is able to optimally remove NOM by itself, numerous integrated processes for the removal of NOM have been studied, such as the combination of photocatalysis with; membrane filtration and adsorption, [93] coagulation, [91] and biodegradation [154].

When focussing on heterogeneous photocatalysis, most research tends to either focus on the optimisation of the photocatalytic activity of $TiO_2$ or to synthesise novel photocatalysts able to compete with $TiO_2$. The improved degradation capabilities of $TiO_2$ are commonly explored via structural modifications (nanocrystals, [155] nanoparticles, [116] nanotubes, [90] nanowires, [117] and nanofilms [118]) and/or combination with other catalysts (ZnO) [156] or materials (polymers, [157] multiwalled carbon nanotubes, [89] and activated carbons [97]). Alternatively, novel photocatalysts are regularly chosen based on their superior photocatalytic activity under near visible or solar light when compared to a $TiO_2$ standard.

Due to large amounts of research focussing on lab scale efficiency, there is an apparent lack of focus on the economics of applying various photocatalytic treatments for the removal of NOM from drinking water sources. This step is crucial to giving a more well-rounded comparison of photocatalytic water treatment with current, well established procedures for NOM removal. Very few publications have evaluated the cost of applying selected photocatalysts for other pollutants, such as immobilised $TiO_2$ for the treatment of industrial wastewaters [158]. Another important factor to consider is the environmental impact of such photocatalytic treatments, life cycle assessments including a goal and scope definition, inventory analysis, and life cycle impact assessment (LCIA) [159] would need to be done to more properly predict the implications of using these systems on an industrial scale.

Owing to the high energy demand of traditional UV-lamps, alternative sources of UV-light are being investigated. One obvious choice of reducing energy demand of UV-light is the use of sunlight for a lower environmentally impacting and cheap light source. The downside of this being that using solar light is typically less effective as an energy source as its emission spectrum has a relatively small overlap with the absorbance of many common photocatalysts, such as $TiO_2$. This is reflected by the large volume of interest in increasing/red shifting the absorbance wavelength range of $TiO_2$ by doping it with different elements such as nitrogen and carbon [113]. Another possible alternative method of UV illumination is the use of LED reactors due to their high efficiency and durability [160,161].

Developing and applying efficient photocatalysis based technologies to remove NOM and mitigate DBP formations is a promising start and making them more efficient and cost-effective for large scale application in integration with other advanced water treatment technologies is the next crucial step to advancing water treatment engineering.

**Author Contributions:** Conceptualization, E.C.; methodology, D.C.A.G. investigation, D.C.A.G.; writing—original draft preparation, D.C.A.G.; writing—review and editing, E.C. and N.R.; supervision, E.C. and N.R. All authors have read and agreed to the published version of the manuscript.

**Funding:** This research was funded by Scottish Water, grant number EP/R513209/1.

**Conflicts of Interest:** The authors declare no conflict of interest.

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
