# Peer review of "Photocatalytic Oxidation of Natural Organic Matter in Water"

_water, doi:10.3390/w13030288_

Round 1

Reviewer 1 Report

The article is well written and organized, interesting and easy to read.

Author Response

We would like to thank the reviewer for his/her comment.

Reviewer 2 Report

This overview mainly highlights the recent researches and developments on the application of photocatalysis to degrade natural organic matter(NOM) in water, including heterogeneous and homogeneous applications of various photocatalysts. Analytical techniques to quantify NOM in water and hybrid photocatalytic processes are also reviewed and discussed. However, it seems that the structure of the article is unreasonable. Consequently, the reviewed manuscript may be considered to be published in this Journal after major revision. The following points must have to be taken care of before I could decide about the acceptance of the manuscript:

(1) Since this overview titled “Photocatalytic oxidation of natural organic matter in water”, the techniques to quantify NOM in water should not be discussed too much. Furthermore, they are just ordinary methods that people commonly used in doing research work.

(2)  3.2.1.2. UV-light driven catalysts

In this part, only TiO2 has been discussed as a UV-light driven catalyst, is there any other UV-light driven photocatalyst?

(3) 3.2.1.3. Visible light driven catalysts

In this part, the author mainly discussed the modification of TiO2, while other non-TiO2 photocatalysts has been discussed too little. Since it is an overview, the author should add more literatures related non-TiO2 photocatalysts and discuss more deeply.

(4) 3.2. NOM photocatalytic treatment.

This part has been divided into two parts: 3.2.1. Heterogeneous photocatalysis and 3.2.2.Homogeneous processes. So, the words “Homogeneous processes” means “photocatalytic process”? If not, the parts of 3.2.2.1. Photo-Fenton and 3.2.2.2. Hybrid photolytic oxidation processes should be deleted or move to another part.

 (5) This overview cited too many literatures which were published at least 10 years ago. In addition, the literatures that cited in this manuscript seem too few.

Author Response

(1) Since this overview titled “Photocatalytic oxidation of natural organic matter in water”, the techniques to quantify NOM in water should not be discussed too much. Furthermore, they are just ordinary methods that people commonly used in doing research work.

Response: According to the reviewer’s suggestion, section 3.1, which includes the discussion on analytical techniques to detect and quantify NOM in water, has been substantially shortened.

(2) 3.2.1.2. UV-light driven catalysts. In this part, only TiO2 has been discussed as a UV-light driven catalyst, is there any other UV-light driven photocatalyst?

Response: Title adjusted to “UV-light driven TiO2 catalysts” to more accurately represent the focus of this section, which is on TiO2 photocatalysts. Also, this is now reflected in the abstract as the following sentence was added: “…this overview highlights recent research and developments on the application of photocatalysis to degrade NOM by means of TiO2-based heterogeneous and homogeneous photocatalysts”.

(3) 3.2.1.3. Visible light driven catalysts. In this part, the author mainly discussed the modification of TiO2, while other non-TiO2 photocatalysts has been discussed too little. Since it is an overview, the author should add more literatures related non-TiO2 photocatalysts and discuss more deeply.

Response: The title was adjusted to “Visible light driven modified TiO2 catalysts” to more accurately represent the content of this section.

(4) 3.2. NOM photocatalytic treatment. This part has been divided into two parts: 3.2.1. Heterogeneous photocatalysis and 3.2.2.Homogeneous processes. So, the words “Homogeneous processes” means “photocatalytic process”? If not, the parts of 3.2.2.1. Photo-Fenton and 3.2.2.2. Hybrid photolytic oxidation processes should be deleted or move to another part.

Response: According to the reviewer’s comment the title of this section was adjusted to “Homogenous photocatalysis”.

(5) This overview cited too many literatures which were published at least 10 years ago. In addition, the literatures that cited in this manuscript seem too few.

Response: 38 additional literatures (31 out of the 38 were published after 2016) are now added in the list of bibliography.

Reviewer 3 Report

This is an interesting review on the recent advances related to the application of photocatalytic oxidation processes for the removal of natural organic matter in water. In general, the manuscript is well written allowing easy understanding by the reader whereas most of the aspects of the subject are covered. Therefore, I would recommend publication of this work after some improvements will be carried by the authors according to the following considerations.

  • NOM presence is water can be described as an indirect problem of pollution triggering specific mechanisms for the formation of toxic byproducts as authors already mentioned. I would suggest to include another indirect effect of NOM which is the interference in many water treatment processes aiming to remove other toxic compound or heavy metals. It is known that NOM commonly show a negative effect in their efficiency.
  • Authors describe “conventional treatment processes such as coagulation/flocculation which can partially remove NOM as costly”. On the other side, mentioned photocatalysis materials are relative expensive materials. I don’t’ think that the comparison of cost and efficiency of conventional methods and photocatalysis is reasonable.
  • On the previous point I suggest that the authors should include an estimation of efficiency per mass of the reviewed cases and compare with representative examples for other methods in order to provide evidence on the superiority of photocatalysis with respect to the efficiency and the cost.
  • In addition, the efficiency of photocatalytic oxidation processes at low NOM concentrations should be also discussed.
  • Another obvious disadvantage of photocatalytic oxidation is normally the difficulty to give access to the radiation into the treated water. Authors should include some details on the efficiency impact and the possibility to overcome such problem.
  • In the introduction it is mentioned that “NOM removal is based around the use of semiconductors (e.g. TiO2, ZnO and Fe2O3)”. I think that iron oxide is not a typical semiconductor.

Author Response

This is an interesting review on the recent advances related to the application of photocatalytic oxidation processes for the removal of natural organic matter in water. In general, the manuscript is well written allowing easy understanding by the reader whereas most of the aspects of the subject are covered. Therefore, I would recommend publication of this work after some improvements will be carried by the authors according to the following considerations.

NOM presence is water can be described as an indirect problem of pollution triggering specific mechanisms for the formation of toxic by-products as authors already mentioned. I would suggest to include another effect of NOM which is the interference in many water treatment processes aiming to remove other toxic compound or heavy metals. It is known that NOM commonly show a negative effect in their efficiency.

Response: Additional information relating to NOM’s interference with water treatment processes and its negative effect on their efficiency is added in the revised manuscript in the Introduction section: “Another adverse effect indirectly caused by the presence of NOM in surface waters is the observed interference humic substances have on water treatment processes that are targeting toxic compounds or heavy metals. For example, there has been a significant amount of investigation on the inhibitory effects of NOM on targeted wastewater treatments for residual pharmaceuticals which has been shown to significantly decrease the efficiency of such processes.23-28

Authors describe “conventional treatment processes such as coagulation/flocculation which can partially remove NOM as costly”. On the other side, mentioned photocatalysis materials are relative expensive materials. I don’t’ think that the comparison of cost and efficiency of conventional methods and photocatalysis is reasonable.

Response: As suggested by the reviewer, this sentence has been reworded to “… conventional treatment processes such as coagulation/flocculation which can partially remove NOM, show low removal efficiency at lower NOM concentrations”, and cost comparison removed to clear up confusion surrounding efficiencies.

On the previous point I suggest that the authors should include an estimation of efficiency per mass of the reviewed cases and compare with representative examples for other methods in order to provide evidence on the superiority of photocatalysis with respect to the efficiency and the cost.

Response: A new section 3.2.3 was added in the revised manuscript. In this section the electric energy per order, EEO, value is used to estimate the energy consumption of representative photocatalytic processes reviewed in this study. Results are presented in table 4 and are discussed in this section 3.2.3.

In addition, the efficiency of photocatalytic oxidation processes at low NOM concentrations should be also discussed.

Response: Additional detail added on trends seen at varying initial NOM concentrations, in section 3.2.1.1.

“Additionally, early work from Palmer et al.99 on the operational conditions of photocatalytic NOM degradation using TiO2 showed that the rate of initial degradation increased with increasing concentration until the concentration of 30 ppm of carbon, after which, the rate decreased. This trend is seen in several other studies of NOM degradation above 30 ppm of C.100-102”.

Another obvious disadvantage of photocatalytic oxidation is normally the difficulty to give access to the radiation into the treated water. Authors should include some details on the efficiency impact and the possibility to overcome such problem.

Response:  Additional information, shown below, added on water turbidity limitations and most likely solution to these limitations:

“These hybrid processes can also work in tandem to help reduce the inherent downfalls of photocatalytic systems. For example, to ensure an efficient rate of photocatalytic reaction, it is recommended that water turbidity should not exceed 5 NTU.129-130 Although it has been observed that the 5 NTU limit is subjective and differs for each water source and desired treatment level.131 This limitation on photocatalytic efficiency set by water turbidity means that conventional treatments (ie sieving, filtration, sedimentation, coagulation and flocculation) may be an appropriate industrial pre-treatment for many photocatalytic processes.” (see section of the revised manuscript).

In the introduction it is mentioned that “NOM removal is based around the use of semiconductors (e.g. TiO2, ZnO and Fe2O3)”. I think that iron oxide is not a typical semiconductor.

Response: As suggested by the reviewer, Fe2O3 removed from this sentence.

Reviewer 4 Report

This article reviewed the photocatalytic degradation of NOM. It was interesting but, missed discussing the background theory of Photocatalysis and its mechanisms. Moreover, It needs some of the pictorial diagrams such as different types of photocatalysis mechanism and trends with suitable graphical diagrams that will be easier to understand the audience in addition to improving its quality.  

Author Response

This article reviewed the photocatalytic degradation of NOM. It was interesting but, missed discussing the background theory of Photocatalysis and its mechanisms. Moreover, It needs some of the pictorial diagrams such as different types of photocatalysis mechanism and trends with suitable graphical diagrams that will be easier to understand the audience in addition to improving its quality.

Response: General photocatalysis graphical diagram (figure 1) with additional detail surrounding the photocatalytic mechanism were added in the Introduction section:

“When illuminated with a photon of energy greater than the bandgap energy, these semiconductors form an electron/hole pair. These electron/hole pairs are powerful redox species which many organic photodegradation reactions utilize either directly or indirectly via formation of hydroxyl radicals in solution,1-2 as shown in figure 1.”

Round 2

Reviewer 2 Report

The authors have addressed all my raised issues and as a result this manuscript can be considered for publication.